# The role of anharmonic phonons in under-barrier spin relaxation of single molecule magnets

Alessandro Lunghi[1,2], Federico Totti[1], Roberta Sessoli[1] & Stefano Sanvito[2]

The use of single molecule magnets in mainstream electronics requires their magnetic moment to be stable over long times. One can achieve such a goal by designing compounds with spin-reversal barriers exceeding room temperature, namely with large uniaxial anisotropies. Such strategy, however, has been defeated by several recent experiments demonstrating under-barrier relaxation at high temperature, a behaviour today unexplained. Here we propose spin–phonon coupling to be responsible for such anomaly. With a combination of electronic structure theory and master equations we show that, in the presence of phonon dissipation, the relevant energy scale for the spin relaxation is given by the lower-lying phonon modes interacting with the local spins. These open a channel for spin reversal at energies lower than that set by the magnetic anisotropy, producing fast under-barrier spin relaxation. Our findings rationalize a significant body of experimental work and suggest a possible strategy for engineering room temperature single molecule magnets.

[1] Università degli Studi di Firenze, Dipartimento di Chimica 'Ugo Schiff', Via della Lastruccia 3-13, Sesto Fiorentino 50019, Italy. [2] School of Physics, AMBER and CRANN, Trinity College, Dublin 2, Ireland. Correspondence and requests for materials should be addressed to S.S. (email: sanvitos@tcd.ie).

Single molecule magnets (SMMs) are molecules comprising a few magnetic ions, which show the properties of both bulk magnets and low-dimensional systems[1]. For instance, SMMs can display magnetic hysteresis together with quantum tunnelling of the magnetization[2]. The magnetism of SMMs can be characterized through relaxation experiments, where an ensemble of molecules is polarized along the direction of an external field and then the magnetization is monitored in time. At high temperature, the relaxation time, $\tau$, exhibits an activated Arrhenius-like behaviour $\tau = \tau_0 e^{U_{eff}/k_B T}$, with $\tau_0$ being the inverse attempt frequency, $k_B$ the Boltzmann constant and $U_{eff}$ the effective barrier for relaxation[1]. Clearly, the use of SMMs in devices, such as non-volatile memories or simply long-living quantum systems[3], requires the relaxation times to be long enough. This in turn means to design molecules with a large $U_{eff}$.

The spin relaxation effective barrier is related to the energies of the spin excited states. For instance, the Hamiltonian of a spin $S$ system with uniaxial anisotropy and zero-field splitting, $D$, is $H = -D\hat{S}_z^2$, so that $U_{eff} = |D|S^2$. One has then two options for enhancing the relaxation barrier, either increasing the molecule total spin or designing compounds with very large magnetic anisotropy (large $D$)[4]. The same argument, of course, applies to any magnetic material and in fact the design of small magnetic bits for data storage is accompanied by using hard magnetic materials. In SMMs, one can use lanthanide ions as magnetic centres and engineer the first coordination shell so to produce a crystal field maximizing the magnetic anisotropy[5]. Such a strategy to enhance $U_{eff}$ has produced some success[6–8], however, several discrepancies remain. While significant deviations from the Arrhenius behaviour at low temperature can be ascribed to tunnelling effects[1,2], there is now a consistent body of evidence showing that the high-temperature spin relaxation follows a thermally activated behaviour with an observed relaxation barrier significantly lower than that measured by, for instance, spectroscopic methods[9–11]. Furthermore, such deviations are sometime more pronounced when $D$ is large[12].

Over-the-barrier activation is essentially a classical diffusive process that requires the system to absorb enough energy to be excited over a potential barrier. In SMMs, such energy is provided by the interaction of the spin with the environment, namely with the molecule vibrations. This is a spin–phonon relaxation mechanism, also known as Orbach relaxation[13]. In a nutshell (see Fig. 1), a phonon with energy $\hbar\omega = U_0$, equal to the difference between the energy of the ground state, $E_0$, and that of the first excited state, $E_2$, is absorbed by the molecule. Such excited state can then relax back to the ground state or to a spin-flipped one of energy $E_1$, which is typically quasi-degenerate with the ground state. The phonon absorbed should have two requirements: to be resonant with the first available spin transition (with the energy barrier in Fig. 1), and to have non-vanishing spin–phonon coupling.

Here we extend the concept of phonon-mediated spin relaxation to the more realistic case in which the phonons acquire a finite lifetime, namely to the situation where the spin couples to an anharmonic phonon bath. We will demonstrate that the finite linewidth of the phonons spectral distribution allows an Orbach-type spin relaxation mechanism even when the phonon is not resonant at the spin excitation energy (see Fig. 1). This has the effect of reducing the effective barrier for relaxation, consistently with the most recent experiments. Such mechanism is first illustrated with a model $S = 1$ spin system, and then calculated with advanced electronic structure theory for a single crystal of $[(\text{tpa}^{Ph})\text{Fe}]^-$ molecules $(S = 2)$, where the ligand $H_3\text{tpa}^{Ph}$ corresponds to the tris((5-phenyl-1H-pyrrol-2-yl)methyl)amine. Our findings introduce new important design elements in the search for slow-relaxing SMMs, namely one has to maximize the

magnetic anisotropy and at the same time engineer the molecule vibrations so to reduce the spin–phonon coupling or the phonon dissipation in the spectral range close to the molecule spin excitations.

## Results

**Spin–phonon dynamics theory.** Let us start by writing down the equations of motion for the coupled spin–phonon system. The low-energy-lying SMM spectrum is composed by a manifold of spin levels, whose degeneracy is removed by relativistic interactions. The total electronic Hamiltonian can then be written as $\hat{H}_0 = \hat{H}_{BO} + \hat{H}_{SO}$, where $\hat{H}_{BO}$ is the non-relativistic Born–Oppenheimer Hamiltonian and all the spin–orbit interactions have been included in $\hat{H}_{SO}$. The ionic degrees of freedom are described by the normal modes of vibration through the Hamiltonian, $\hat{H}_{ph} = \sum_\alpha \hbar\omega_\alpha(\hat{n}_\alpha + 1/2)$, where $\hat{n}_\alpha = \hat{a}_\alpha^\dagger \hat{a}_\alpha$ is the phonon density operator and $\hat{a}_\alpha^\dagger$ $(\hat{a}_\alpha)$ is the creation (annihilation) operator for a phonon of frequency $\omega_\alpha$. Molecular vibrations, either within a single molecule or through the relative motion of the molecules in a crystal, modulate the spin–orbit interaction. This results in a spin–phonon coupling Hamiltonian that in first approximation is linear in the ionic displacement, $\hat{q}_\alpha = \frac{1}{\sqrt{2}}\left(\hat{a}_\alpha^\dagger + \hat{a}_\alpha\right)$, and can be written as

$$\hat{H}_{s-ph} = \sum_\alpha \left(\frac{\partial\hat{H}_0}{\partial\hat{q}_\alpha}\right)_0 \hat{q}_\alpha. \tag{1}$$

The dynamics of the entire system (spin plus phonons) follow the time evolution of the total density operator, $\hat{\rho}(t)$,

$$\frac{d\hat{\rho}(t)}{dt} = \frac{i}{\hbar}\left[\hat{\rho}(t), \hat{H}\right], \tag{2}$$

where $\hat{H} = \hat{H}_0 + \hat{H}_{ph} + \hat{H}_{s-ph}$ is the total Hamiltonian. Equation (2) can be simplified by assuming that the phonon dynamics is much faster than the spin one, as it is found for slow-relaxing SMMs. The Born–Markov approximation allows us to integrate out the phonons' component of the density matrix and reduces the problem to a purely electronic one in the presence of a phonon bath. The dynamics of the spin degrees of freedom can then be studied through the first-order-reduced spin density operator, $\hat{\rho}^S$, as described by the diagonal elements of the Redfield equation[14]

$$\frac{d\rho_{aa}^S(t)}{dt} = \frac{2}{\hbar^2}\sum_c \sum_\alpha \mathcal{M}_{ac}^\alpha \rho_{cc}^s(t),$$
$$\mathcal{M}_{ac}^\alpha = -\sum_j V_{aj}^\alpha V_{jc}^\alpha \delta_{ac} G(\omega_{jc}, \omega_\alpha) + |V_{ac}^\alpha|^2 G(\omega_{ca}, \omega_\alpha), \tag{3}$$

where $\rho_{ab}^S = \langle a|\hat{\rho}^S|b\rangle$, $V_{ab}^\alpha = \langle a|\frac{\partial\hat{H}_0}{\partial\hat{q}_\alpha}|b\rangle$, $|a\rangle$ is the eigenfunction of the spin system described by the Hamiltonian $H_0$ with energy $E_a$ and $\omega_{ab} = (E_a - E_b)/\hbar$. Thus in equation (3) $|V_{ac}^\alpha|^2 G(\omega_{ca}, \omega_\alpha)$ represents the kinetic rate of population transfer between the eigenstates $|a\rangle$ and $|c\rangle$.

The spectral representation of the Green's function for the phonon bath, $G(\omega_{ij}, \omega_\alpha)$, is the central quantity of our discussion, since it contains the temperature dependence of the spin dynamics. A rigorous treatment of this quantity would require knowledge of the phonons dissipation process, namely of the anharmonic constants. This means solving the appropriate phonons equations of motion or approximating the relevant self-energy if a perturbative approach is possible. Until now, only the zeroth-order term of $G$ has been introduced into the theory[1,13]. This corresponds to the 'undamped' case, that is, to the harmonic approximation to bath dynamics. Here we make a step forward and we explicitly introduce phonons dissipation by modelling their spectral shape. Thus, $G$ of each phonon assumes a

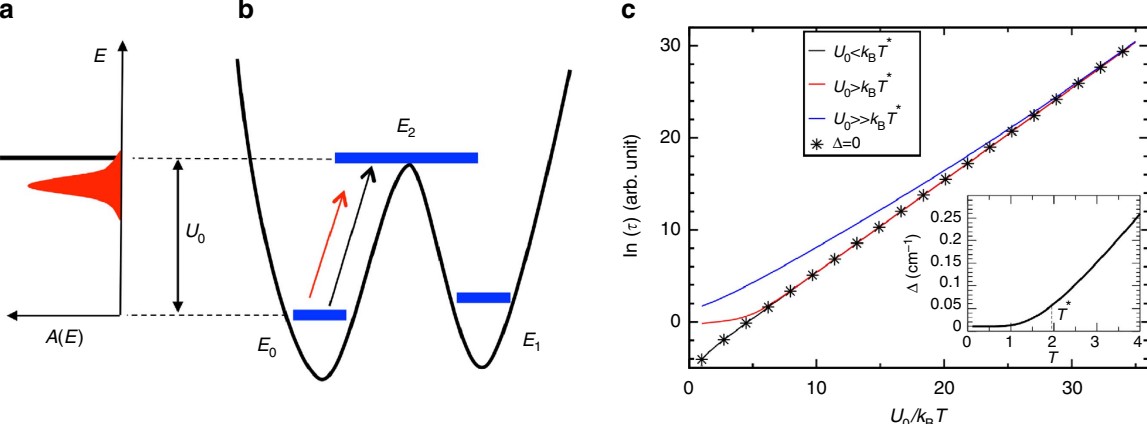

**Figure 1 | Phonon-induced spin relaxation for an $S = 1$ system.** (**a**) Displays a Lorentzian and a Dirac-like phonons' density of states. The spin energy barrier profile is pictured in **b**, where the ground state of energy $E_0$ is separated from the spin-flip state of energy $E_1$ by a barrier $U_0 = E_2 - E_0$. The spin relaxation occurs by exciting the spin system to the state of energy $E_2$ via absorption of a phonon. In the standard Orbach's process, the phonon is resonant with the spin excitation energy and its spectral function, $A(E)$, is a $\delta$-function (black bar in **a**). In contrast, when one considers a finite phonon lifetime (red curve in **a**), the phonon does not need to be resonant with the spin levels. In **c**, we report the logarithm of the relaxation time $\tau$ against the temperature $T$ scaled by the excitation energy $U_0/k_B$. The inset reports the qualitative behaviour of the phonon linewidth, $\Delta$, as function of the temperature, where $T^\star$ represents the temperature above which the anharmonic effects start to be important. The black symbols describe the Arrhenius behaviour expected from the standard Orbach process, while the solid lines represent the expected behaviour for anharmonic crystals in three different regimes.

Lorentzian shape with amplitude $\Delta_\alpha$,

$$G\left(\omega_{ij}, \omega_\alpha\right) = \frac{\Delta_\alpha \bar{n}_\alpha}{\Delta_\alpha^2 + \left(\omega_{ij} - \omega_\alpha\right)^2} + \frac{\Delta_\alpha\left(\bar{n}_\alpha + 1\right)}{\Delta_\alpha^2 + \left(\omega_{ij} + \omega_\alpha\right)^2}, \quad (4)$$

where $\bar{n}_\alpha = \frac{1}{e^{\beta\hbar\omega_\alpha} - 1}$ is the mean phonon occupation number and $\beta = 1/k_B T$. Note that the lifetime of the $\alpha$-th mode is $\tau_\alpha = \hbar/\Delta_\alpha$.

The inset of Fig. 1c schematically illustrates the typical temperature dependence of $\Delta_\alpha$, which takes a small constant value for $T < T^\star$ and then becomes linear above the threshold temperature, $T^\star$ (refs 15–17). The spin dynamics then qualitatively depends on the value of $U_0$ with respect to $k_B T^\star$ (see Fig. 1c). For $U_0 \ll k_B T^\star$, the phonons have a long lifetime ($\Delta_\alpha$ is small) over the entire temperature range in which the spin dynamics is usually measured ($k_B T < U_0$). Therefore, at any relevant temperature, spin relaxation takes place through the standard Orbach's mechanism for harmonic phonons, with $\tau$ following an Arrhenius-like activated temperature dependence and $U_{\text{eff}} = U_0$. This is a frequent situation found in low-anisotropy SMMs, where indeed the first excitation energy can be associated with the effective barrier for relaxation. In contrast, for $U_0 > k_B T^\star$, the temperature dependence of $\Delta_\alpha$ becomes important. At high temperature, there are significant deviations from a single-exponent Arrhenius behaviour, which can no longer describe relaxation across the entire temperature range. Importantly, if one insists in fitting the experimentally measured relaxation curve at high temperature with an Arrhenius plot, an effective barrier significantly lower than $U_0$ will be estimated (see Fig. 1c, for $U_0 \gg k_B T^\star$). This situation becomes relevant for SMMs with high anisotropy and already explains why one often measures $U_{\text{eff}} \ll U_0$.

**$S = 1$ embedded in a stochastic anharmonic phonons bath.** A simple $S = 1$ model will help us in discussing the microscopic mechanism behind spin relaxation in the presence of inelastic phonons. Since calculating $\Delta_\alpha$ would require an out-of-reach molecular dynamics simulations (from several ps to ns) for a crystal containing hundreds of atoms, we have taken here a stochastic approach. According to Kubo's model[18], the phonons line-shape corresponds to the amplitude of the Gaussian probability distribution of the $\alpha$-mode's energy fluctuations. In

the canonical ensemble (NVT), this quantity is evaluated as

$$\Delta_\alpha^2 = \frac{\partial\left\langle H_{\text{ph}}^\alpha\right\rangle_{\text{NVT}}}{\partial\beta} = \frac{\left(\hbar\omega_\alpha\right)^2 e^{\beta\hbar\omega_\alpha}}{\left(e^{\beta\hbar\omega_\alpha} - 1\right)^2}, \quad (5)$$

with $H_{\text{ph}}^\alpha = \hbar\omega_\alpha(\hat{n}_\alpha + 1/2)$. Such temperature dependence of $\Delta_\alpha^2$ is in qualitative agreement with both the experimental and theoretical homogeneous linewidth temperature behaviour[15,17,19]. This choice of $\Delta$ corresponds to the case where the phonons rapidly become over damped as $T$ increases and it can be considered as opposite to the harmonic limit.

Consider now the three-level system of Fig. 1b, where the states $|0\rangle$, $|1\rangle$ and $|2\rangle$ are, respectively, the ground state, its spin-reversed state and the excited state. The degeneracy between $|0\rangle$ and $|1\rangle$ is assumed to be slightly lifted for instance by a magnetic field. Assume also that there is only one phonon of frequency $\omega$ that can couple to the spin transition $i \rightarrow j$ with strength $V_{ij}$. Such phonon is then coupled anharmonically to a phonon's bath so to acquire a finite linewidth $\Delta$. Note that this model applies to the $S = 1$ case and also to any situation with a low-energy-lying almost degenerate doublet separated by an excited state. If one solves equation (3) for the three-level system in the harmonic limit ($\Delta \rightarrow 0$) (see Supplementary Note 2), the relaxation time will be $\tau = \tau_0 e^{\beta\hbar\omega}\delta_{\hbar\omega, U_0} = (1/V_{02})e^{\beta\hbar\omega}\delta_{\hbar\omega, U_0}$, that is, relaxation will proceed through an Arrhenius behaviour with $U_{\text{eff}} = U_0$ and a pre-exponential factor inversely proportional to the spin–phonon coupling coefficient $V$. Importantly, in this limit one requires the phonon to be resonant at the excitation energy $U_0$. Note that the same situation holds true for small $U_0$ and $U_0 \gg k_B T$, that is, when $T$ is sufficiently low that $\Delta$ is practically temperature independent ($T < T^\star$).

In contrast, for an anharmonic phonon with linewidth described by equation (5), the relaxation time for $\hbar\omega > k_B T$ is calculated as in equation (38) of Supplementary Note 2:

$$\tau = \frac{\hbar\omega}{V_{02}}\left[e^{\frac{\beta\hbar\omega}{2}} + \frac{(U_0 - \hbar\omega)^2}{(\hbar\omega)^2}e^{\frac{3}{2}\beta\hbar\omega}\right]. \quad (6)$$

As expected, there is no relaxation, $\tau \rightarrow \infty$, if the spin–phonon coupling vanishes, since no other mechanism is considered here (for example, quantum tunnelling). A much more surprising fact

is that, for $V_{02} \neq 0$, the relaxation time follows an activated temperature dependence, but the effective barrier is uniquely determined by the phonon frequency. In particular, when the phonon is resonant with the spin level, we find $\tau = U_0/V_{02} e^{\frac{U_0}{2k_B T}}$, meaning that relaxation follows an Arrhenius behaviour with the effective barrier being half of the excitation energy $U_{eff} = U_0/2$. Note that the harmonic picture cannot be recovered from equation (6), due to the assumptions made in its derivation (see Supplementary Note 2). However, in this case of anharmonic phonons, and in stark contrast with the harmonic case often discussed in literature[13,20,21], the resonance between the phonon energy and the splitting of the spin levels is not a necessary condition for the relaxation, which can take place even off resonance with the more complex temperature dependence described by equation (6).

The same off-resonance mechanism described above applies to any other spin transition. For instance, see Supplementary equation (29) and Supplementary equation (30) of Supplementary Note 2. If $V_{02} = 0$, relaxation can still take place by a phonon-induced off-resonant transition between $|0\rangle$ and $|1\rangle$. This gives a relaxation time

$$\tau = \frac{\hbar\omega}{V_{01}} e^{\frac{\beta\hbar\omega}{2}}, \qquad (7)$$

which again depends on the phonon frequency alone. Note, however, that such 'direct' relaxation mechanism, at variance with the Orbach's one, is usually observed at very low temperature where the condition $T > T^\star$ is hardly achievable and a closer agreement with the harmonic scenario is therefore expected. Clearly, if both $V_{01}$ and $V_{02}$ do not vanish, there will be competition between the different relaxation channels and in general $\tau$ will take a more complex form. Yet, our analysis clearly demonstrates that in addition to the anisotropy barrier, also the phonon frequency and the phonon lifetime become a relevant energy scale of the problem. In some cases, it is the dominant one. Our mechanism goes beyond previous attempts at explaining the sub-barrier relaxation measured in some SMMs[22–24], where a phenomenologically finite linewidth was added to the spin excited state energy levels. Such early approaches, in fact, although able to explain $U_{eff} < U_0$, fail in relating the spin relaxation process to a physically sound dissipation mechanism. Furthermore, large barrier reductions require massive linewidths, in stark contrast to those measured spectroscopically. Indeed, a typical experimental electron paramagnetic resonance linewidth for SMMs is of the order of $1\,\text{cm}^{-1}$ or less[25].

It should be stressed that the effects due to the phonon linewidth broadening induced by the phonon–phonon interactions are effective regardless the nature of the density of states of the lattice vibrations. Indeed, even a single phonon interacting with the spin is sufficient to induce a reduction of $U_{eff}$. However, the broadening of the phonons' spectral shape also enables more phonons, close in energy, to be operative at the same time, making the vibrational density of states another important figure of merit for the interpretation of the spin dynamics.

***Ab initio* spin dynamics**. A full quantitative analysis of our mechanism now requires the calculation of the phonon spectrum and the spin–phonon coupling for all possible modes and all possible spin transitions. We then abandon simple models for an *ab initio* description, and investigate the magnetization dynamics of the SMM $[(\text{tpa}^{Ph})\text{Fe}]^-$ (ref. 25). This SMM exhibits a $S = 2$ ground state with large uniaxial anisotropy (see Fig. 2a) and shows slow relaxation rates in an external applied magnetic field[25]. This specific SMM has been chosen due to its well experimentally characterized structural and magnetic properties.

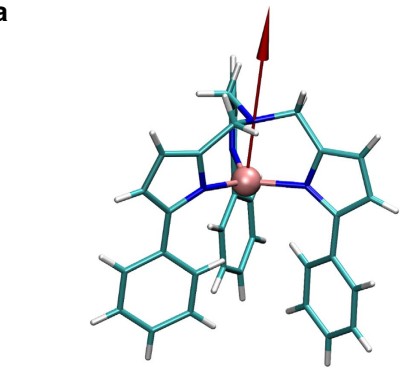

**a**

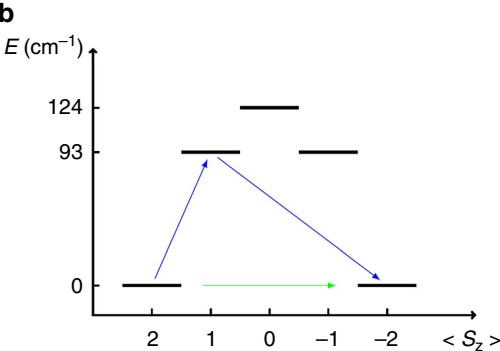

**b**

**Figure 2 | The $S = 2$ $[(\text{tpa}^{Ph})\text{Fe}]^-$ SMM.** In **a**, we show the optimized molecular structure and in **b** the corresponding energy diagram. Atoms colour code: Fe = pink, N = blue, C = green and H = white. The red arrow lying along the pseudo $C_3$ molecular symmetry axis shows the magnetization easy axis direction. The energy diagram presents five spin states and it has been calculated with the molecule oriented with its easy axis along the external magnetic field. The first four states are only slightly non-degenerate ($\sim 0.5$–$1\,\text{cm}^{-1}$) and their energy difference could not be appreciated in the figure. The green arrow represents the direct relaxation pathway and the blue arrow represents the Orbach's relaxation mechanism.

The relaxation features showed by $[(\text{tpa}^{Ph})\text{Fe}]^-$ are rather general and, given the large magnetic anisotropy ($D = -27.5\,\text{cm}^{-1}$ (ref. 25)), they readily translate to most appealing single ion SMMs, comprising either a transition metal or a lanthanide ion. Moreover, this SMM has one of the most compact unit cell in the SMMs family, which makes it easier the implementation of our demanding computational framework. The $[(\text{tpa}^{Ph})\text{Fe}]^-$ crystal has $P\bar{1}$ symmetry and the unit cell contains two SMMs and two $\text{Na}^+$ cations, coordinated by three dimethoxyethanes. Since $[(\text{tpa}^{Ph})\text{Fe}]^-$ possesses an orbital non-degenerate ground state, we can separate the electronic and pure spin degrees of freedom, and describe it with an effective spin Hamiltonian, $\hat{H}_S = \sum_{ij} D_{ij}\hat{S}_i\hat{S}_j$, where $\hat{S}_i$ is a cartesian component of the spin operator[26]. We simulate the $[(\text{tpa}^{Ph})\text{Fe}]^-$ spectrum with the multi-determinant wave-function active space SCF scheme (CASSCF)[27,28]. The $2S + 1 = 5$ lowest-lying $S = 2$ CASSCF energy roots have been used to fit the spin Hamiltonian $\hat{H}_S$, as provided by the Orca package (see 'Methods' section, and Supplementary Note 4 for a discussion on the validity of the spin Hamiltonian approach). The resulting energy ladder is reported in Fig. 2b, where a 1500 Oe external magnetic field has been introduced through a Zeeman term $\hat{H} = g_e\beta\vec{B}\cdot\vec{S}$ ($g_e\beta$ is the gyromagnetic ratio). The external field is here introduced merely to be consistent with experimental conditions, optimized to reduce tunnelling relaxation, particularly efficient in zero field. All the matrix elements derivatives that appear in equation (3) can be written in terms of $\hat{H}_S$ (see

Supplementary Note 3) as

$$\sum_{\alpha} \langle S_b | \frac{\partial \hat{H}_0}{\partial \hat{q}_\alpha} | S_a \rangle = \sum_{\alpha} \sum_{ij} \frac{\partial D_{ij}}{\partial \hat{q}_\alpha} \langle S_b | \hat{S}_i \hat{S}_j | S_a \rangle, \qquad (8)$$

where $|S_n\rangle$ is an eigenstate of $\hat{H}_S$. The right-hand side expression in equation (8) makes it possible to implement spin–phonon coupling calculations through numerical differentiation of the spin Hamiltonian coefficients $D_{ij}$, which in turn are calculated by CASSCF method.

All the spin–phonon models employed to date to study the dynamics of SMMs collapse the phonon spectrum in a single mode (the Debye model) so to obtain an handy relation for the relaxation time[1], and only recently the validity of this approximation has been discussed[29]. Here the Debye assumption is no longer required as all spin–phonon coupling coefficients are calculated, meaning that all the potential spin relaxation channels are taken into account. The only limitation in our case is practical and it is dictated by the size of the system that we can simulate. We consider a periodic crystal and calculate only the unit cell gamma-point normal modes. These consist of all short wavelength intra-molecular modes and the limited number of inter-molecular ones compatible with having two molecules per unit cell.

Next, we solve the master equation, equation (3), in time for different temperatures, finding an exponential decay of the magnetization at any $T$. The relaxation time is then extracted by fitting the time evolution of the magnetization component parallel to the external magnetic field with the function $M_z(t) = (M_z(0) - M_z(\infty))e^{-t/\tau} + M_z(\infty)$. The results of our calculations are reported as black dots in Fig. 3, where we present the logarithm of $\tau$ as a function of the inverse temperature $1/T$.

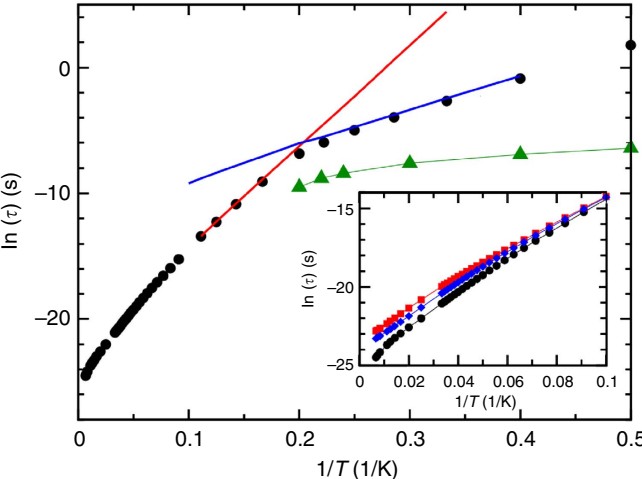

**Figure 3 | [(tpa$^{Ph}$)Fe]$^-$ ab initio spin dynamics results.** Calculated temperature dependence of the relaxation time, $\tau$. Black dots are for simulations where all the relaxation processes are considered. The green triangles and the green line represent the experimental results as taken from ref. 25. The blue and red lines represent calculations performed, respectively, by considering only transitions between the quasi-degenerate doublet, and by completely neglecting them (that is, by considering only relaxation processes activated through the excited states). Arrhenius's fits performed for the $T < 5$ K and for $5$ K $< T < 10$ K cases return effective barriers of 19.7 cm$^{-1}$ and 55.7 cm$^{-1}$, respectively. In the inset, we show the relaxation time in the high-$T$ range calculated using only a limited number of phonon modes when all the relaxation processes are included. The red line with squares describes the simulation done with only five modes, the blue line with diamonds corresponds to the simulation done with 15 modes and the black line with dots corresponds to the full-phonon spectra case.

Clearly, there is no single exponential dependence of $\tau$ over the entire temperature range, indicating that non-resonant spin–phonon relaxation channels contribute to the dynamics, as discussed for the simple $S = 1$ case. In particular, we notice that $\ln(\tau)$ is approximately linear with $1/T$ for $T < 5$ K ($1/T > 0.2$) and then displays a strong deviation. The linear slope extracted for $T < 5$ K gives us $U_{eff} = 19.7$ cm$^{-1}$, which is in excellent agreement with the experimental value, $U_{eff} = 26$ cm$^{-1}$, measured in a similar temperature range[25]. Notably, such value is much smaller than our calculated excitation energy of 94 cm$^{-1}$ (corresponding to the first excited two-fold degenerate state with $\langle \hat{S}_z \rangle = \pm 1$, see Fig. 2b), demonstrating that phonon-induced under-barrier relaxation can drastically reduce the spin lifetime of a SMM.

To isolate the dominant relaxation mechanism at different temperatures, we perform the same exercise done for the simple $S = 1$ model, namely we solve the master equation first by retaining only the transitions between the almost degenerate ground state doublet (blue curve in Fig. 3), and then by neglecting these and considering only the remaining ones. For $T < 5$ K non-resonant intra-doublet transitions dominate completely the dynamics and provide the largest contribution to $\tau$. An Arrhenius fit of the blue curve essentially returns the same value, $U_{eff} = 19.7$ cm$^{-1}$, obtained when fitting the complete one for $T < 5$ K. Although several phonon modes may contribute to $\tau$, we observe that $U_{eff} = 19.7$ cm$^{-1}$ is remarkably close to half of the energy of the lowest phonon mode ($\sim 36$ cm$^{-1}$). This makes us concluding that the low-$T$ relaxation of [(tpa$^{Ph}$)Fe]$^-$ is dominated by direct doublet transitions via the lowest phonon mode.

In contrast, when only phonon-activated transitions through the excited states are considered, we obtain the red curve of Fig. 2, which follows closely the complete curve in the intermediate range $5$ K $< T < 10$ K and can be fitted with an Arrhenius plot and $U_{eff} = 55.7$ cm$^{-1}$. Thus, in this intermediate $T$ range, the relaxation is driven by non-resonant phonon-activated processes. These contribute to the barrier with $\frac{3}{2}\hbar\omega$, again suggesting that it is the $\hbar\omega = 36$ cm$^{-1}$ mode to dominate the dynamics. This normal mode of vibration is complex in nature being a delocalized representation of a large number of unit cell degrees of freedom. Finally, when the temperature is further enhanced (this is a regime never investigated experimentally since the $\tau$s are too short) several modes participates to the dynamics and a single Arrhenius fit is no longer possible. A numerical demonstration of such multi-mode relaxation process is provided in the inset of Fig. 3 where we show calculations including all the transitions and an increasingly large number of phonon modes. Notably the relaxation times get smaller as more modes are available, meaning that more activated relaxation channels become possible.

## Discussion

Finally, let us now critically discuss the results just obtained. In general, we expect the physics shown here for [(tpa$^{Ph}$)Fe]$^-$ to be quite common in high-anisotropy SMMs. The most recurrent experimental observation is that of a single-exponent Arrhenius-like relaxation, with an activation barrier significantly smaller than that expected from the molecule excitation spectrum. Here we have clearly shown that identifying $U_{eff}$ with $U_0$ is no longer justified, since $U_{eff}$ is also determined by the specific molecule phonon structure and by the spin–phonons coupling. In fact, we have demonstrated that off-resonance phonon modes always contribute to lower $U_{eff}$ with respect to $U_0$, as observed experimentally.

Intriguingly, in literature there are also reports of SMMs displaying different relaxation regimes at different temperatures. For instance, the low-$T$ relaxation time of Dy$^{3+}$ is defined by a double-exponent Arrhenius curve with the two fitted effective

barriers (334 and 94 K) being lower than the first accessible spin exited state (430 K)[30]. This behaviour suggests the existence of two different spin relaxation mechanisms which, in light of our discussion, could be attributed to the effect of different normal modes. In fact, we believe that the most common situation is the one presented here, namely the one in which the relaxation time can be associated to different barriers at different temperatures. In experiments, the temperature range is usually narrow, so that a single-exponent Arrhenius behaviour is often observed. Notably under-barrier relaxation has been recently explained on the basis of a second-order Raman effect[9,31,32], but here we clearly show that one does not necessarily need to claim second-order effects to re-conciliate theory with experiments.

Besides the temperature-dependent component, the magnitude of $\tau$ is also defined by a pre-exponential coefficient (see equation (6)), $\tau_0$, which in turn depends on the matrix elements of the spin–phonon coupling Hamiltonian and the actual phonon frequencies. This complex dependence over the system details makes it difficult to predict the relaxation time on the sole basis of the molecule excitation spectrum and indeed calls for a more complete treatment, going also beyond the Debye model[29,33]. Nevertheless, a few general designing rules for increasing $U_{eff}$ may be provided by our analysis. For instance, the direct relaxation between the two quasi-degenerate ground states can be slowed down by either increasing the smallest phonon frequency and by reducing the spin–phonon coupling coefficients. The first strategy translates in designing more structurally rigid SMMs, while the second one is readily accomplished with the usual quantum tunnelling reduction methods such as by employing highly axial symmetry Kramer ions.

At high temperature, the goal is instead that of quenching the Orbach's mechanism. Although complicated to exploit, a possible strategy may be that of modulating the spin excited state energy to obtain a non-resonant condition with the phonon spectrum. In this context, interesting solutions may be offered by 3D and on surface 2D crystal engineering techniques[34,35]. However, the selective engineering of the spin–phonon coupling coefficients could be the most effective way to slow down the Orbach's relaxation rate and more detailed studies will be needed. Our finding are in line with the recent research trend that see the engineering of SMM/phonon spectra as a major challenge for the design of the next-generation magnetic molecular materials[9,32,36–38]. Here we have shown that the vibrational features of SMMs are the central figures of merit in the design of new systems. Indeed, frequency magnitude and lifetime of phonons, along with the spin–phonon coupling coefficients, are all quantities that directly enter in the spin equation of motions and strongly affect the relaxation timescales.

In conclusion, we have here recast the basic concepts of spin–phonon relaxation in a fully quantum mechanical formalism. This eliminates previously adopted approximations aiming at simplifying the problem and provides a general theoretical framework amenable to *ab initio* methods. Our approach requires the calculation of the full-phonon spectrum, the spin Hamiltonian and the spin–phonon coupling, all quantities that enter a master equation for the spin dynamics. We have shown that the phonons dissipation, treated here at a stochastic level, enables off-resonance spin relaxation, whose magnitude and temperature dependence are determined by the electronic and vibrational details of the specific SMM. A fully quantitative analysis for $[(tpa^{Ph})Fe]^-$ demonstrates that the relaxation time has a complex temperature behaviour, where different relaxation mechanisms dominate over different energy ranges. Our work suggests several designing strategies for reducing the spin relaxation and thus for engineering long-living SMMs.

Furthermore, we remark that spin–phonon interaction is common to any magnetic material and the theory outlined here does not apply just to SMMs. Moreover, as other classes of magnets show a long magnetization lifetime, it is possible to speculate that they could also be affected by similar spin-bath dynamical features as those reported here.

## Methods

**Density functional theory calculations.** All the structural optimization and Hessian calculations have been performed with the CP2K software[39] at the level of density functional theory with the Perdew-Burke-Ernzerhof (PBE) functional including Grimme's D3 van der Waals corrections[40,41]. Other dispersion forces correction schemes have been tested and compared with each other in describing the unit cell structure. Among the available df-D, df-D2, rvv10 and D3 methods, Grimme's one has been found to be the most accurate[42]. A double-zeta polarized MOLOPT basis set and a 600 Ry of plane-wave cutoff have been used for all the atomic species. The same set-up has been used to evaluate the Γ-point phonons of the entire periodic unit cell. A finite differences algorithm for the evaluation of the Hessian matrix with a 0.001 a.u. differentiation step has been used.

***Ab initio* magnetic properties calculations.** The ORCA software[43] has been employed for computing all the magnetic properties. Calculations of the **D** anisotropy tensor have been performed at the CASSCF level with a def2-TZVP basis set for Fe and N, def2-SVP for C and H and a def2-TZVP/C auxiliary basis set for all the elements. This choice has been carefully tested and it correctly reproduces both spin–orbit corrected spectra and the **D** derivatives calculated by employing the def2-TZVP basis set for all the atomic species. A (6,5) active space have been chosen as recommended in literature and the spin–orbit contribution has been included through quasi-degenerate perturbation theory. Although CASSCF calculations could be done only on isolated molecules, we take into account electrostatic effects by embedding the single $[(tpa^{Ph})Fe]^-$ molecule in a $21 \times 21 \times 21$ lattice of RESP point charges[44] mimicking the periodicity of the crystal. Although the crystal is made of charged molecules, its $P_{\bar{1}}$ imposes a quenching of the electrostatic fields and indeed the inclusion of the point charges does not produce any sizable effect.

***Ab initio* mapping of the spin Hamiltonian.** Due to the presence of low-lying excited electronic states, the spin Hamiltonian formalism is here only partially correct and it could not exactly fit the entire lower-lying spectrum. Other parameterizations have been tested, including for instance the fourth-order $\hat{O}_{4q}$ operators in the fit, but, although the agreement with the CASSCF results improve (RSS passes from 689 to $6.0 \times 10^{-9}$), it does not change the results that we have discussed. To evaluate all the $V_{ab}^\alpha$ coefficients in equation (3), we numerically derived the **D** anisotropy tensor by modulating the equilibrium molecular structure of one $[(tpa^{Ph})Fe]^-$ molecule along all normal modes by $\pm 0.2$ units.

**Data availability.** All the relevant data discussed in the present paper are available from the authors on request.

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

# ARTICLE

12. Fataftah, M. S., Zadrozny, J. M., Rogers, D. M. & Freedman, D. E. A mononuclear transition metal single-molecule magnet in a nuclear spin-free ligand environment. *Inorg. Chem.* **53,** 10716–10721 (2014).

13. Orbach, R. Spin-lattice relaxation in rare-earth salts. *Proc. R. Soc. A* **264,** 458–484 (1961).

14. Breuer, H. P. & Petruccione, F. *The Theory of Open Quantum Systems* (Oxford University Press, 2002).

15. Bini, R., Califano, S., Eckert, B. & Jodl, H. J. Temperature dependence of the vibrational relaxation processes in natural and isotopically pure 32S8: effect of the isotopic impurities on infrared phonon lifetimes. *J. Chem. Phys.* **106,** 511–518 (1997).

16. Pinan, J. P., Ouillon, R., Ranson, P., Becucci, M. & Califano, S. High resolution Raman study of phonon and vibron bandwidths in isotopically pure and natural benzene crystal. *J. Chem. Phys.* **109,** 5469–5480 (1998).

17. Deinzer, G., Birner, G. & Strauch, D. *Ab initio* calculation of the linewidth of various phonon modes in germanium and silicon. *Phys. Rev. B* **67,** 144304–144306 (2003).

18. Kubo, R. Stochastic Liouville equations. *J. Math. Phys.* **4,** 174–183 (1963).

19. Torre, R., Righini, R., Angeloni, L. & Califano, S. Picosecond measurements of relaxation of internal modes in crystalline benzene as a function of temperature. *J. Chem. Phys.* **93,** 2967–2973 (1990).

20. Abragam, A. & Bleaney, B. *Electron Paramagnetic Resonance of Transition Ions* (Oxford University Press, 2012).

21. Shrivastava, K. N. Theory of spin-lattice relaxation. *Phys. Status Solidi* **117,** 437–458 (1983).

22. Gill, J. A theory of the 'phonon bottleneck' in the Orbach process of spin-lattice relaxation. *J. Phys. C* **6,** 109–120 (1973).

23. Lyo, S. K. Interference and intermediate-level-width corrections to the Orbach relaxation rate. *Phys. Rev. B* **5,** 795–802 (1972).

24. Young, B. A. & Stapleton, H. J. Apparent lowering of energy levels as measured by Orbach relaxation rates. *Phys. Lett.* **21,** 498–501 (1966).

25. Harman, W. H. *et al.* Slow magnetic relaxation in a family of trigonal pyramidal iron (II) pyrrolide complexes. *J. Am. Chem. Soc.* **132,** 18115–18126 (2010).

26. Tennant, W. C. & Walsby, C. J. Rotation matrix elements and further decomposition functions of two-vector tesseral spherical tensor operators; their uses in electron paramagnetic resonance spectroscopy. *J. Phys. Condens. Matter* **12,** 9481–9495 (2000).

27. Atanasov, M., Ganyushin, D., Pantazis, D. A., Sivalingam, K. & Neese, F. Detailed *ab initio* first-principles study of the magnetic anisotropy in a family of trigonal pyramidal iron (II) pyrrolide complexes. *Inorg. Chem.* **50,** 7460–7477 (2011).

28. Atanasov, M. *et al.* First principles approach to the electronic structure, magnetic anisotropy and spin relaxation in mononuclear 3D-transition metal single molecule magnets. *Coord. Chem. Rev.* **289–290,** 177–214 (2015).

29. Hoffmann, S. K. & Lijewski, S. Raman electron spin-lattice relaxation with the Debye-type and with real phonon spectra in crystals. *J. Magn. Reson.* **227,** 51–56 (2013).

30. Watanabe, A., Yamashita, A., Nakano, M., Yamamura, T. & Kajiwara, T. Multi-path magnetic relaxation of mono-dysprosium(III) single-molecule magnet with extremely high barrier. *Chem. Eur. J.* **17,** 7428–7432 (2011).

31. Novikov, V. V. *et al.* A trigonal prismatic mononuclear cobalt(II) complex showing single-molecule magnet behavior. *J. Am. Chem. Soc.* **137,** 9792–9795 (2015).

32. Rechkemmer, Y. *et al.* A four-coordinate cobalt(II) single-ion magnet with coercivity and a very high energy barrier. *Nat. Commun.* **7,** 10467 (2016).

33. Hoffmann, S. K. & Lijewski, S. Phonon spectrum, electron spin-lattice relaxation and spin-phonon coupling of $Cu^{2+}$ ions in $BaF_2$ crystal. *J. Magn. Reson.* **252,** 49–54 (2015).

34. Basagni, A. *et al.* Stereoselective pho-topolymerization of tetraphenylporphyrin derivatives on Ag(110) at the sub-monolayer level. *Chem. Eur. J.* **20,** 14296–14304 (2014).

35. Braga, D. & Grepioni, F. Intermolecular interactions in nonorganic crystal engineering. *Acc. Chem. Res.* **33,** 601–608 (2000).

36. Gomez-Coca, S. *et al.* Origin of slow magnetic relaxation in Kramers ions with non-uniaxial anisotropy. *Nat. Commun.* **5,** 4300 (2014).

37. Escalera Moreno, L., Suaud, N. & Arino, A. G. Theoretical determination of the spin-vibration coupling in the highly coherent molecular spin qubit $[Cu(mnt)_2]^{2-}$. Preprint at http://arxiv.org/abs/1512.05690 (2016).

38. Ganzhorn, M., Klyatskaya, S., Ruben, M. & Werns-dorfer, W. Strong spin-orbit coupling between a single-molecule magnet and a carbon nanotube nanoelectrome-chanical system. *Nat. Nanotechnol.* **8,** 165–169 (2013).

39. Hutter, J., Iannuzzi, M., Schiffmann, F. & VandeVondele, J. Cp2k: atomistic simulations of condensed matter systems. *Wiley Interdiscip. Rev. Comput. Mol. Sci.* **4,** 15–25 (2014).

40. Perdew, J. P., Burke, K. & Wang, Y. Generalized gradient approximation for the exchange-correlation hole of a many-electron system. *Phys. Rev. B* **54,** 533–539 (1996).

41. Grimme, S., Antony, J., Ehrlich, S. & Krieg, H. A consistent and accurate *Ab initio* parametriza-tion of density functional dispersion correction (DFT-D) for the 94 elements H-Pu. *J. Chem. Phys.* **132,** 154104–154119 (2010).

42. Tran, F. & Hutter, J. Nonlocal van der Waals functionals: the case of rare-gas dimers and solids. *J. Chem. Phys.* **138,** 204103–204109 (2013).

43. Neese, F. The ORCA program system. *Wiley Interdiscip. Rev. Comput. Mol. Sci.* **2,** 73–78 (2012).

44. Bayly, C. I., Cieplak, P., Cornell, W. & Kollman, P. A. A well-behaved electrostatic potential based method using charge restraints for deriving atomic charges: the RESP model. *J. Phys. Chem.* **97,** 10269–10280 (1993).

## Acknowledgements

This work was supported by the European Research Council Quest and Advanced Grant MolNanoMaS (no. 267746) projects. We also acknowledge the MOLSPIN COST action CA15128. Computational resources were provided by the Trinity Centre for High Performance Computing (TCHPC) and the Irish Centre for High-End Computing (ICHEC). A special acknowledgement is due to Prof Roberto Righini from the University of Florence and LENS for the many useful discussions.

## Author contributions

All the authors conceived the project, analysed/discussed the results and wrote the manuscript. A.L. performed all the calculations and developed the theory jointly with S.S.

## Additional information

**Competing financial interests:** The authors declare no competing financial interests.

