## [Peer Review File · Nature Communications]

Reviewers' comments:

Reviewer #1 (Remarks to the Author):

The authors present a theoretical work, where they couple spin and phonon degrees of freedom to explain spin relaxation of single molecule magnets. Assuming finite lifetimes for the phonon the Orbach relaxation mechanism is used to explain spin relaxation of single molecule magnets with apparent barriers smaller than expected. The finite lifetime is assumed to be due to anharmonicity. The phonon-phonon interaction is not treated explicitly but included in a stochastic way based on the assumption that the interaction drives also thermal equilibrium.

I find the manuscript well written and the arguments are presented in a very logical and accessible way.

There is convincing evidence that the described process may be important to understand the spin relaxation behavior in single molecule magnets.

However, I would like to ask the authors to make comments to the following points:

A) Finite lifetimes have been introduced in the past to describe 'phonon bottleneck' [see e.g. Gill (1973): A theory of the 'phonon bottleneck' in the Orbach process of spin-lattice relaxation J. Phys. C: Solid State Phys. 6 109] which normally increases relaxation time. The principal model appears to be very similar (see figure 1).

B) Connected with the previous point. Is the assumption of thermal equilibrium in case of SMM really fulfilled? [see e.g. Gill (1975): The establishment of thermal equilibrium in paramagnetic crystals Rep. Prog. Phys. 38 91]

I think that this point warrants some short discussion because it appears to be required for the stochastic treatment.

C) While it may be true that all models used to study the dynamics of single molecule magnets collapse the phonon spectrum in a single mode, there is at least one example where the spin-phonon matrix elements have been calculated in a similar manner as in the present work however only at the DFT level. [see Pederson (2002): Fourth-order magnetic anisotropy and tunnel splittings in Mn₁₂ from spin-orbit-vibron interactions Physical Review Letters 89 (9), 097202] There the spin-phonon interaction is used to calculate higher order terms of the Hamiltonian. It seems that in the present treatment the spin Hamiltonian remains unchanged by the vibrations of the molecule but couples only with a bath of phonons.

D) In case of more complicated systems like e.g. Fe₈ the spin multiplett S=10 overlaps with other spin multipletts, so that cross relaxation may play a role. What is the importance of other relaxation mechanisms?

E) Just a remark; The Orbach relaxation can be described as a second order Raman process. For that reason I do not really agree with the formulation on page 6, line 413.

Reviewer #2 (Remarks to the Author):

This manuscript is recommended for publication in Nature Communication subject to revisions. Report is being attached as pdf.

Reviewer #3 (Remarks to the Author):

The paper by Sanvito and co-workers detail the spin-phonon coupling for the case of non-discrete phonon density of states. In particular the paper addresses the importance of this case for rationalizing thermally assisted relaxation processes for molecular magnets with apparent barriers below the spectroscopically determined barriers. This contribution is very important. It is also very timely, as it clarifies a field which recently has become increasingly messy by uncritical reporting of barriers for thermally assisted relaxation of magnetization without attention to the underlying physics.

Considering the fairly heavy theoretical core material in the paper, the exposition is very pedagogical and successfully targets the chemistry-based molecular magnetism community. However, I think that several readers, like myself, will have some difficulty in figuring out if the phonon dispersion necessarily has to originate in anharmonicity. Or put in another way: is "anharmonic phonons..." in the text and title really the optimal wording if chemists should get the full message. Could "phonon dispersion" or "continuous phonon spectra.." substitute for the anharmonicity. I think this point should be discussed briefly no matter the answer. In doing this, the title should also be reconsidered.

I was a bit surprised to find that one of the most closely related papers (Luis et. al. Nature Comm. 5: 4300) discussing photon-assisted magnetic relaxation processes - albeit in a more narrow context - is not referenced or placed into the context of the present contribution at all. I think this should be done. It will detract absolutely nothing from the beautiful presentation in question, but will probably aid a large fraction of the readers.

The paper illustrates the theoretical description nicely by application to a concrete example, [t_{ps}PhFe]-. However, the reader is presented with absolutely no arguments for this particular choice of test system. I think a rationale for this choice would strengthen the paper and avoid speculations that the system was chosen based on agreement with the modeling.

In line 204: "phonons" or "phonon" ? Also the sentence commencing on line 445 is rather inelegant due to the insistence on the plural.

I think the sentence ending on line 268 deserves at least one reference.

Several other recent contributions have commented on the importance of controlling/tailoring the phonon spectra of SMM crystals, these ought to be referenced in the concluding part (line 442 and onward).

Report on submission NCOMMS-16-20231

This work reports a new mechanism for spin relaxation in so-called single molecule/ single ion magnets (SMM or SIM) where low-frequency phonon states operative in the temperature range 2-10 K (typical for relaxation times measured using alternating current susceptometry) take the role of an activation barrier, usually identified with the energy gap U_0 between the ground and lowest excited magnetic sublevels. Using master equations and advanced electronic structure theory calculations the authors demonstrate, that one-phonon under-barrier relaxation is able to account for effective activation barriers U_{eff} deduced from Arrhenius τ vs $1/T$ plots much lower than U_0 . This is in contrast with usual interpretations of $U_{\text{eff}} < U_0$ in terms of a two-phonon Raman mechanism. This work also shows the inadequacy of the Debye model (widely used in the community) for SMM and SIMs with high magnetic anisotropy (zero-field splitting D). The authors define also ranges of parameters (U_0 , T^* - onset of phonon anharmonicity and T where the effect is predicted to occur ($T^* < T < U_0$) and discuss under what conditions this mechanism could be suppressed (weak spin-phonon coupling, $\hbar\omega_{\text{phonon}} > U_0$, $T < T^*$) in favor of larger spin relaxation times. I consider this work very important; it will influence for the field of molecular magnetism and I can recommend it for publication in Nature Communications.

Yet there are some points that should be addressed prior to publishing of this work.

- 1) Using the key equations 3 to 5 implies, as stated on page 2 that the phonon dynamics is much faster than the spin-relaxation rate. However, for phonons of large energy the model cannot be applied, since according to eq.5 phonon lifetimes get much larger than usual spin relaxation times. For example, according to eq.5 for a phonon with frequency 100 cm^{-1} at $T=2\text{K}$ one gets $\tau = 1397 \text{ s}$ which is orders of magnitude larger than the spin-relaxation time (0.01 s) in a system like Fe(tpaPh) (Ref.23, Figure S6). It then follows, that out of the many modes only limited number of modes, those that obey $T^* < U_0/k_B$ at temperatures $T > T^*$ can be used to estimate the effect, modes with larger energies have to be excluded from the very beginning from the consideration; for the same reason one has to exclude very low frequency modes where τ_{phonon} tends to become zero (again within the limits of the model).
- 2) Along the same lines and within the approximations inherent as stated in Supp.Inf. only modes with final Δ_a will contribute to the spin relaxation; at very low temperatures a non-zero value of Δ_a has to be assumed in order to apply the model.
- 3) Eqs.4 have been derived in Supp.Inf. I, but eqs.6 and 7 not (“it is possible to demonstrate”); when is it possible, under what conditions?
- 4) The statement that Fe(tpaPh) is in an orbitally non-degenerate ground state (page 4, line 283) is not correct; the complex possesses an 5E orbitally degenerate ground state which is prone to Jahn-Teller distortions. This complicates the problem somewhat. In the Supporting information this problem is qualitatively addressed but to avoid any ambiguity these explanations should be shifted to the main part. It may set some limits to the conclusions made. Static Jahn-Teller distortions are quenched by spin-orbit coupling leading to a A_1, A_2 relativistic ground state (C_{3v}^* double group notations) but the spin-Hamiltonian is in a strict sense not well defined here. Yet, the authors use a second order spin-Hamiltonian.
- 5) The authors **are the first** to apply advanced electronic structure methods in getting spin-vibronic parameters which is gratifying. One therefore expects getting more details from the Supporting information (III) about the procedure and numerical

results (values) for: optimized geometries for the 228 atom cluster from Bulk and Isolated Models, vibrational frequencies, first and second derivatives of D_{ij} . The latter would prove that linear spin-phonon coupling parameters are obtained from the linear term in a quadratic dependence of D_{ij} on each and every normal mode.

- 6) The authors certainly have not computed spin-phonon coupling parameters **for all modes** as declared in the insert of Figure 3 in the main text states, but only few low frequency one that yield finite phonon live times. Which are the phonons involved and what is the shape of the lowest phonon mode of 36 cm^{-1} and the first and second derivatives of D_{ij} with respect to this mode? The latter will be responsible for a two-phonon Raman processes. This would stimulate further work aiming at comparing the two mechanisms.
- 7) Section V of the Supporting information is difficult if not impossible to read. This part does not really communicate properly with the body of the paper: **i)** the authors use another, more general form (eqs.43-46) compared to eq.3. Which of the two equations have been employed for the simulations of Figures 1, 3 (main text) and Figure 4(Supp.Inf); can the authors list more explicit expressions for the phonon correlation functions appearing in 43-46 as was been done in Section I of the supporting information? **ii)** Eigenstates of S_z (Figure 4) makes only sense if an external d.c. field is applied (here 1500 Oe). But its effect is already included in the eigenvectors, correct? It is then not clear why the blue and red lines in Figure 4 differ from each other. **iii)** The text on page 12 top (Supporting information) discusses phase dynamics in S_z analog to the tunneling effect; the latter was eliminated by applying a d.c. magnetic field of 1500 Oe, correct? Furthermore relaxation times turn out to be effected by the static field as pointed out on experimental grounds elsewhere (Chem.Sci.2013,4,125-138, Figure 4), this is not in the model. **iv)** Figure 4 has been constructed taking some unspecified model parameters in the calculation; this, as already mentioned makes the text (page 12) difficult to read and follow; why is M_z on Figure 4 (read curve) not starting from 2.00? **v)** In the discussion of this part the oscillating a.c. magnetic field should be explicitly taken into consideration, this is again not in the model. **vi)** Finally, it appears from the text between lines 224 and 228 on page 12, that the authors used the discussion in the upper part to justify why only diagonal terms in eq.3 in the main part have been considered. As far as it not clear what the authors mean in the preceding discussion and how to quantify the arguments this is of no help for understanding of the assumptions of the model.
- 8) What parameters have been used for the construction of Figure 1(c)? From the inset I deduce a phonon frequency of 27.5 cm^{-1} . Having identified the model parameters used to construct Figure 1(c) one can also avoid "arbitrary units" and put real computed spin relaxation times.
- 9) Temperature dependence of the spin-relaxation time for Fe(tpaph) complex can be deduced from the maxima in the variable-frequency out-of-phase ac susceptibility, Figure S6 of Ref.23. It would be of help for the sake of comparison having experimental relaxation times along with the computed ones in one plot (Figure 3, main text).

I. REVIEWER 1

Referee: The authors present a theoretical work, where they couple spin and phonon degrees of freedom to explain spin relaxation of single molecule magnets. Assuming finite lifetimes for the phonon the Orbach relaxation mechanism is used to explain spin relaxation of single molecule magnets with apparent barriers smaller than expected. The finite lifetime is assumed to be due to anharmonicity. The phonon-phonon interaction is not treated explicitly but included in a stochastic way based on the assumption that the interaction drives also thermal equilibrium.

I find the manuscript well written and the arguments are presented in a very logical and accessible way. There is convincing evidence that the described process may be important to understand the spin relaxation behavior in single molecule magnets.

Reply: We thank the referee for her/his positive comments about our work.

Referee: A) Finite lifetimes have been introduced in the past to describe 'phonon bottleneck' [see.e.g. Gill (1973): A theory of the 'phonon bottleneck' in the Orbach process of spin-lattice relaxation *J. Phys. C: Solid State Phys.* 6 109] which normally increases relaxation time. The principal model appears to be very similar (see figure 1).

Reply: We thank the referee for pointing out this interesting publication. In our manuscript we have already cited two similar attempts to explain effective barrier reductions through the introduction of a spin excited state line-width. However, although closely related to our findings, that approach has a major conceptual issue. Both in our formalism and in that used by the former contributors to the field, the phonon life-time is the only meaningful quantity, as phonons dynamics is considered independent from and not affected by the spin-dynamics. The spin dynamics, instead, is treated explicitly. In this way the inclusion of a spin lifetime in the spin equation of motion would violate the causality principle. Indeed the lifetime considered in our work is the phonon one and not that of the spin. Moreover, and this is a central point of the discussion, our shift of focus from the spin details to the phonons ones is crucial for the interpretation of the spin dynamics itself and serves as a fundamental new guideline for the design of new SMMs.

Changes to the manuscript: The discussion above is already present in the manuscript. In response to the referee we have added the reference to the fundamental work of Gills (ref [21]) to the other two already present. The paragraph at page 4 (first column) now reads:

“Our mechanism goes beyond previous attempts at explaining the sub-barrier relaxation measured in some SMMs [21, 22, 23], where a phenomenologically finite line-width was added to the spin excited-state energy levels. Such early approaches, in fact, although able to explain $U_{\text{eff}} < U_0$, fail in relating the spin-relaxation process to a physically-sound dissipation mechanism. Furthermore large barrier reductions require massive line-widths, in stark contrast to those measured spectroscopically. Indeed, a typical experimental EPR line-width for SMMs is of the order of 1 cm^{-1} or less [24].”

Referee: B) Connected with the previous point. Is the assumption of thermal equilibrium in case of SMM really fulfilled? [see e.g. Gill (1975): The establishment of thermal equilibrium in paramagnetic crystals *Rep. Prog. Phys.* 38 91] I think that this point warrants some short discussion because it appears to be required for the stochastic treatment.

Reply: We thank the referee for raising such a relevant point. The importance of the non-equilibrium properties of the lattice has been recognized many times in literature and also by some of us very recently in a paper concerning a $S = 1/2$ molecular magnet (“Giant spin-phonon bottleneck effects in evaporable vanadyl-based molecules with long spin coherence”, *Dalton Transactions*, DOI:10.1039/C6DT02559E, 2016). However, the spin-phonon bottleneck is more often encountered in spin-half systems, where the direct relaxation process occurs through sparsely dense regions of the crystal density of states. As also pointed in the references cited by the referee, Orbach relaxation, which is at the centre of our discussion, is only marginally affected by the spin-phonon bottleneck. As such, we are confident that the assumption of having an equilibrated phonons population is rather realistic. Indeed, the spin lifetime is in the μs -ms range for the investigated temperatures, which is more than enough for the vibrational system to thermalize. For instance see the references [5] and [6] cited in the Supplementary Information (ESI), where the experimental relaxation time of molecular phonons is measured in the ns-ps window, in full support of our stochastic model.

Changes to the manuscript: In order to remark this point we have added a comment at the end of section I of the ESI:

“By using Δ_α as defined in Eq. (15), the life-time of the normal mode $\omega=36 \text{ cm}^{-1}$ is in the $ns \div ps$ range for temperatures in the $2 \div 10 \text{ K}$ interval. This is orders of magnitude shorter than the spin relaxation time observed for $[(tpa^{Ph})Fe]^{-1}$ ($ms \div \mu s$), demonstrating that Markov’s approximation is valid for the dynamics discussed in the main text of the paper. When the Markov’s condition is not fulfilled, one has then to replace Eq. (21) with the more general Eq. (17).”

Referee: While it may be true that all models used to study the dynamics of single molecule magnets collapse the phonon spectrum in a single mode, there is at least one example where the spin-phonon matrix elements have been calculated in a similar manner as in the present work however only at the DFT level. [see Pederson (2002): Fourth-order magnetic anisotropy and tunnel splittings in Mn12 from spin-orbit-vibron interactions Physical Review Letters 89 (9), 097202]. There the spin-phonon interaction is used to calculate higher order terms of the Hamiltonian. It seems that in the present treatment the spin Hamiltonian remains unchanged by the vibrations of the molecule but couples only with a bath of phonons.

Reply: Indeed the referee is right to bring to our attention the contribution of Pederson and co-workers. In the paper mentioned by the referee the phonons of the system are considered as a perturbation to the spin Hamiltonian and they are used to explain the modification of the spin level energies observed in experiments. In our work we consider a completely different scenario, where the phonons act as dynamical source of energy dissipation for the spin instead of being a static perturbation, which induces only an average energy renormalization. Such energy level renormalization also arises naturally in our framework when considering the second order derivatives of the spin Hamiltonian with respect to a single normal mode. As also stated by the authors of Pederson’s paper, the effect originating from the second order derivatives is more important than the one explicitly discussed by them in the paper.

Changes to the manuscript: We have added in section IV of the SI a discussion of this effect for the normal mode at 36 cm^{-1} , which is the vibration inducing the relaxation in the system studied here:

“The determination of the second order derivatives of \mathbf{D} with respect to a single normal mode q_α also offers the possibility to calculate the average effect of the phonon bath dynamics on the spin Hamiltonian. This effect has been proposed by Pederson et al. [18] and very recently by Moreno et al.[19] to be at the origin of spin Hamiltonian parameters renormalization effects. Assuming the spin-phonon coupling Hamiltonian to be

$$H_{sph} = \sum_{\alpha} \left[\frac{\partial H_s}{\partial q_\alpha} + \frac{1}{2} \frac{\partial^2 H_s}{\partial q_\alpha^2} \right],$$

it is possible to evaluate the renormalization of the spin Hamiltonian parameters due to the average effect of the bath defining a new \bar{H}_s as

$$\bar{H}_s = H_s + Tr_B[H_{sph}\rho_B]$$

Only the second order derivatives of the spin Hamiltonian contribute to the trace over the bath degrees of freedom appearing in Eq. (48) leading to

$$\bar{H}_s = H_s + \sum_{\alpha} \frac{\partial^2 H_s}{\partial q_\alpha^2} (\bar{n}_\alpha + \frac{1}{2})$$

According to Table IV and Eq. (49) it is possible to estimate a phonon induced modification of the D_{ij} elements of the order of fraction of cm^{-1} for $\omega = 36 \text{ cm}^{-1}$ in the investigated temperature range. This is clearly only a negligible correction to the effects discussed in the paper and can not be assumed to be at the origin of the massive spin-flip energy barrier reductions observed in literature.”

As discussed in SI this renormalization effect represents only a minor correction to the parameters calculated here.

Referee: In case of more complicated systems like e.g. Fe8 the spin multiplett S=10 overlaps with other spin multipletts, so that cross relaxation may play a role. What is the importance of other relaxation mechanisms?

Reply: The formalism employed in this work is completely general and can be applied to any SMMs, regardless of their spin structure. In the specific case of multi-ion SMMs the more practical way to describe the relaxation would be through a multi-spin Hamiltonian, where all the single ion anisotropies and all the exchange couplings (either isotropic

and anisotropic) among the spins are explicitly included. This approach prevents any possible flaws associated with the giant spin approximation and automatically include all inter- and intra-multiplets relaxation pathways. We thank the referee for her/his interest in our work and for raising such interesting question. We will address this issue in future research, where other classes of SMMs will be considered.

Referee: Just a remark; The Orbach relaxation can be described as a second order Raman process. For that reason I do not really agree with the formulation on page 6, line 413.

Reply: We are afraid but this time we have to disagree with the referee; the Orbach and the Raman process are two different relaxation mechanisms. It is true that the Orbach mechanism concerns the absorption of a phonon and the instantaneously emission of another one. This, however, must not be confused with a second order Raman process. Indeed, the former one comes from a combination of two first order processes (the absorption and emission of a phonon) with an intermediate state corresponding to a population state of the spin density matrix operator. The occupation number of this population state of the density matrix remains zero due the instantaneously nature of the process. In this respect, the Orbach mechanism is the spin equivalent of an optical resonant fluorescence process. This point is addressed in section II of the SI when discussing the Orbach solution of the $S = 1$ model. In contrast, the Raman process involves the creation of coherence states of the density matrix only involving the excited state, due to higher order Feynman diagram in the time-dependent perturbation theory, or two-phonon absorption/emission. It must also be stressed that the difference in these two processes is substantial as their intensity is due to different features of the system and, as a consequence, their temperature dependence is different as well. For the details about the Raman process, not addressed here, the referee may refer to K.N. Shrivastava, “Theory of Spin-Lattice Relaxation, Phys. status solidi 117, 437–458 (1983)”

II. REVIEWER 2

Referee: This work reports a new mechanism for spin relaxation in so-called single molecule/ single ion magnets (SMM or SIM) I consider this work very important; it will influence for the field of molecular magnetism and I can recommend it for publication in Nature Communications.

Reply: We thank the referee for her/his very thorough review of our manuscript and for the encouraging words concerning the possible impact of our work. We respond here to all the comments raised.

Referee: Using the key equations 3 to 5 implies, as stated on page 2 that the phonon dynamics is much faster than the spin-relaxation rate. However, for phonons of large energy the model cannot be applied, since according to eq.5 phonon lifetimes get much larger than usual spin relaxation times. For example, according to eq.5 for a phonon with frequency 100 cm^{-1} at $T=2\text{K}$ one gets $\tau = 1397 \text{ s}$ which is orders of magnitude larger than the spin-relaxation time (0.01 s) in a system like Fe(tpaPh) (Ref.23, Figure S6). It then follows, that out of the many modes only limited number of modes, those that obey $T^* < U_0/k_B$ at temperatures $T > T^*$ can be used to estimate the effect, modes with larger energies have to be excluded from the very beginning from the consideration; for the same reason one has to exclude very low frequency modes where τ_{phonon} tends to become zero (again within the limits of the model).

Reply: This is an important point and the referee is indeed right: Markov's approximation can be applied only for phonons with a certain lifetime. This is certainly one of the major drawbacks of the previous theories of spin-phonon coupling, where one was considering at the same time both an infinite lifetime for the phonon (harmonic approximation) and Markovian equations of motion. We have reformulated section I of the SI in the hope of offering to the reader a more clear overview of our new model, highlighting its own strengths and limitations. In this new section we derive Markov's approximation for the spin-reduced density matrix equations of motion only at the end of the discussion. In doing so the limitations of our approach become more transparent. We also provide the reader with a non-Markovian equation of motion to be used in case Markov's condition is not fulfilled.

Changes to the manuscript: Section I of the SI has been considerably re-elaborated to include the just mentioned discussion. For the referee's convenience, we remand to the main SI text, where new parts have been marked in blue. We would like to stress the fact that the dynamics discussed in the main body of the text is perfectly valid under Markov's approximation and we believe that the discussion of possible non-markovian effects in the magnetization dynamics of SMMs is beyond the scope of the paper. To this regards we here would like to report only the new concluding part of the new section I of the SI:

"By using Δ_α as defined in Eq. (15), the life-time of the normal mode $\omega=36 \text{ cm}^{-1}$ is in the ns÷ps range for temperatures in the $2 \div 10 \text{ K}$ interval. This is orders of magnitude shorter than the spin relaxation time observed for $[(tpa^{Ph})Fe]^-$ (ms÷mus), demonstrating that Markov's approximation is valid for the dynamics discussed in the main text of the paper. When the Markov's condition is not fulfilled, one has then to replace Eq. (21) with the more general Eq. (17)."

Referee: Along the same lines and within the approximations inherent as stated in Supp.Inf. only modes with final Δ_α will contribute to the spin relaxation; at very low temperatures a non-zero value of Δ_α has to be assumed in order to apply the model.

Reply: Please refer to the previous answer and to the response to the 2nd question of Reviewer 1.

Referee: Eqs.4 have been derived in Supp. Inf. I, but eqs.6 and 7 not (it is possible to demonstrate); when is it possible, under what conditions?

Reply: Eqs. 6 and 7 in the main text are demonstrated in section 2 of the SI and a reminder to the supplementary information was already present in the main text. For sake of clarity we have now added a more explicit reminder in the main text.

Changes to the manuscript: We have added the following section just before Eq. 6:

"In contrast, for an anharmonic phonon with line-width described by Eq. (5), the relaxation time for $\hbar\omega > k_B T$ is

calculated as in Eq. (38) of section II of the SI.”

Referee: The statement that Fe(tpaPh) is in an orbitally non-degenerate ground state (page 4, line 283) is not correct; the complex possesses an 5E orbitally degenerate ground state which is prone to Jahn-Teller distortions. This complicates the problem somewhat. In the Supporting information this problem is qualitatively addressed but to avoid any ambiguity these explanations should be shifted to the main part. It may set some limits to the conclusions made. Static Jahn-Teller distortions are quenched by spin-orbit coupling leading to a A1, A2 relativistic ground state (C3v* double group notations) but the spin-Hamiltonian is in a strict sense not well defined here. Yet, the authors use a second order spin-Hamiltonian.

Reply: As stated by the referee, a short discussion on the validity of the spin Hamiltonian was already presented in sec. IV of the SI. In order to make the point more clear we have extended the discussion in such section. Please refer to the text in the SI. Here we would like to emphasize that the use of the spin Hamiltonian formalism, although not mandatory, was somehow required given the status of the literature. We have in fact noticed that a well-defined notion of spin-phonon coupling was sometimes missing. As such we believe that a proper formulation of the spin-phonon coupling in term of spin Hamiltonian derivatives is extremely useful to clarify the physical origin of this quantity in the mathematical language used by a very large community.

Changes to the manuscript: We have added the following discussion to Section IV of the SI.

“Before proceeding to the discussion of the **D** tensor calculations, a comment on the validity of the spin Hamiltonian itself is required. In a completely general fashion, the spin Hamiltonian formalism can be applied to describe the splitting of the spin energy levels as long as the wave-functions of the electronic states considered [see Eq. (43)] possesses a well-defined S^2 expectation value, *i.e.* there is no spin contamination. Technically this becomes an issue only after the inclusion of the spin-orbit interaction, which may lead to a ground state multiplet with non-vanishing orbital angular momentum. In general Jahn-Teller activity removes the problem, as in the present case, where the global C_3 molecular symmetry is relaxed to a lower C_1 symmetry, as can also be seen from Table I. However, the stiffness of the first coordination shell makes the Jahn-Teller distortions not fully effective in removing the degeneracy of the first two excited states and the final low-energy-lying spin-orbit-corrected wave-functions are indeed a superposition of the two different eigenstates of the Born-Oppenheimer Hamiltonian. However, in this case the two solutions have both an $S=2$ multiplicity and, therefore, this situation does not impose any formal restriction on the use of the spin Hamiltonian. It must also be stressed that the spin Hamiltonian formalism has also been already used before to study $[(tpa^{Ph})Fe]^-$ and other synthetic analogs [11, 17]. Moreover, it is worth mentioning that our framework does not necessarily involve the use of the spin Hamiltonian. Indeed, one can directly use the expression (43) to evaluate the matrix elements in the first term of Eq. (45), instead of passing by the fitting of the spin Hamiltonian.

Furthermore, we have inserted a pointer to such discussion in the main text.

“The $2(S + 1) = 5$ lowest-lying $S = 2$ CASSCF energy roots have been used to fit the spin Hamiltonian HS, as provided by the Orca package (see Methods, and Section IV of the SI for a discussion on the validity of the spin Hamiltonian approach).”

Referee: The authors are the first to apply advanced electronic structure methods in getting spin-vibronic parameters which is gratifying. One therefore expects getting more details from the Supporting information (III) about the procedure and numerical results (values) for: optimized geometries for the 228 atom cluster from Bulk and Isol Models, vibrational frequencies, first and second derivatives of D_{ij} . The latter would prove that linear spin-phonon coupling parameters are obtained from the linear term in a quadratic dependence of D_{ij} on each normal mode.

Reply: The referee is right, our paper describes a rather laborious calculation and not many details have been provided. We have amended such lack of computational details in section IV of the SI. In particular we have now introduced: 1) details of the optimized geometries for both the Isol and Bulk models; 2) details of the lattice parameters optimization for the Bulk Model; 3) extension of the discussion about the spin Hamiltonian validity (see above); 4) details about the numerical differentiation of the spin Hamiltonian parameters; 5) the effect of the spin Hamiltonian parameters due to the averaged static effect of the phonon bath. The vibrational density of states for both the Isol and Bulk models were already present in SI. We also would like to mention that we effectively perform the numerical differentiation of the D tensor for ALL the gamma point normal modes. However, their detailed discussion will be

presented in a following paper as it is beyond the scope of the present one.

Changes to the manuscript: Section IV of the SI has now been considerably extended to include most of the computational details. These include significant more text (in blue for the convenience of the referee) and two new figures.

Referee: The authors certainly have not computed spin-phonon coupling parameters for all modes as declared in the insert of Figure 3 in the main text states, but only few low frequency one that yield finite phonon live times. Which are the phonons involved and what is the shape of the lowest phonon mode of 36 cm^{-1} and the first and second derivatives of D_{ij} with respect to this mode? The latter will be responsible for a two-phonon Raman processes. This would stimulate further work aiming at comparing the two mechanisms.

Reply/Changes to the manuscript: We have reported the values of first and second order derivatives for all the \mathbf{D} tensor components with respect to the normal mode at 36 cm^{-1} . We also added to the text their graphical representation for visual inspection. Concerning the nature of the mode at 36 cm^{-1} we have added the following sentence to the main text:

”This normal mode of vibration is complex in nature being a delocalized representation of a large number of unit-cell degrees of freedom.”

The discussion of the Raman relaxation would require the knowledge of all the mixed second order derivatives with respect to the normal modes. The number of possible derivatives is huge and they have not been calculated here (note that the quantity and quality of calculations required for our proposed spin-relaxation mechanism is already very substantial). Such a discussion is, at the moment, beyond the scope of the present paper but it will be certainly addressed in a future work.

Referee: Section V of the Supporting information is difficult if not impossible to read. This part does not really communicate properly with the body of the paper: i) the authors use another, more general form (eqs.43-46) compared to eq.3. Which of the two equations have been employed for the simulations of Figures 1, 3 (main text) and Figure 4 (Supp.Inf); can the authors list more explicit expressions for the phonon correlation functions appearing in 43-46 as was been done in Section I of the supporting information? ii) Eigenstates of S_z (Figure 4) makes only sense if an external d.c. field is applied (here 1500 Oe). But its effect is already included in the eigenvectors, correct? It is then not clear why the blue and red lines in Figure 4 differ from each other. iii) The text on page 12 top (Supporting information) discusses phase dynamics in S_z analog to the tunneling effect; the latter was eliminated by applying a d.c. magnetic field of 1500 Oe, correct? Furthermore relaxation times turn out to be effected by the static field as pointed out on experimental grounds elsewhere (Chem.Sci.2013,4,125-138, Figure 4) , this is not in the model. iv) Figure 4 has been constructed taking some unspecified model parameters in the calculation; this, as already mentioned makes the text (page 12) difficult to read and follow; why is M_z on Figure 4 (read curve) not starting from 2.00 ? v) In the discussion of this part the oscillating a.c. magnetic field should be explicitly taken into consideration, this is again not in the model. vi) Finally, it appears from the text between lines 224 and 228 on page 12, that the authors used the discussion in the upper part to justify why only diagonal terms in eq.3 in the main part have been considered. As far as it not clear what the authors mean in the preceding discussion and how to quantify the arguments this is of no help for understanding of the assumptions of the model.

Reply/Changes to the Manuscript: The referee is right in her/his comment and we have now consistently re-elaborated section V of the SI and we remand the referee to the text of the ESI for its new complete final version. Regarding the referee’s concerns (we followed the referee numbering starting from point vi, since it is somehow overarching other comments):

vi) This section has two main goals, to define the computational procedure used for the simulation of the magnetization dynamics and to justify the use of only the diagonal part of the Redfield equation. In order to make these two points more clear we have added at the beginning of the section a discussion explaining in more details how we have simulated the $\ln(\tau)$ vs $1/T$ plot in the main text by starting from the diagonal part of the Redfield equation. Only later we have discussed the effect of using the full Redfield equation. We have removed the final discussion about the pure dephasing time T_2^* as it is not necessary for our discussion.

i) By re-elaborating the section we have also adopted a formalism more coherent with the rest of the paper. We now explicitly state which equation has been used to produce the results.

ii) In our simulation the choice of the starting state is rather arbitrary and we believe that both S_z and the eigenstates of H_S are meaningful and worth to be discussed. In order to make a more clear connection to experiments we have also added two comments about possible strategies to effectively prepare H_S eigenstates:

”One can visualize this in silico experiment as a preparation of the quantum state by thermalizing the system at a temperature where only the ground state is populated and it is afterward quenched by abruptly increasing the temperature to the desired value.”

and

”This scenario can be obtained in experiments by a preparation of the quantum state in an extremely high external B field followed by a quench due to the reduction of B to the desired value.”.

iii) The quantum tunneling mentioned by the referee is the so-called ”quantum tunneling of the magnetization”, which is a relaxation mechanism. This process is indeed quenched by external field as it proceed through a direct population transfer between a two-fold degenerate ground state induced by the presence of inter-SMM interactions of magnetic dipolar origin. The external B field would remove the degeneracy of the ground-state doublet reducing the relaxation rate. This process is not included in our discussion as we considered just one SMM. The ultra-fast fluctuations observed in the S_z eigenstate dynamics are instead the Rabi oscillations of the phase. We have added the following sentence to make this point more clear:

”This effect must not be confused with the so called quantum tunnelling relaxation, which is instead a direct relaxation mechanism between two degenerate HS eigenstates due to inter molecule dipolar interactions. This mechanism is not covered by our simulations as only one SMM has been considered.”

The effect of the external magnetic field is fully included in our formalism and accounted beyond perturbation theory by diagonalizing the Zeeman interaction at the same time of the other terms in the spin Hamiltonian. Indeed, by changing the intensity of B the two-fold degeneracy of the HS eigenstates is progressively removed. However, as clearly stated in literature, the external magnetic field B would effect only the direct relaxation mechanism and the tunneling of the magnetization (the latter is not included in our models) as they are directly related to the degeneracy of the HS eigenkets. B only marginally affects the Orbach process, at the centre of our discussion.

iv) The starting value of the red line is not 2 because of the presence of a rhombic term in the spin Hamiltonian that mix the S_z components in the H_S eigenstates. We have added the following sentence to the text:

”The expectation value of M_z at $t = 0$ is slightly lower than 2 due to the presence of the rhombic term in H_S , which mixes the S_z components of the H_S eigenkets and lowers the magnetization expectation value. This effect is reduced by the presence of the external magnetic field (1500 Oe) present in the spin Hamiltonian but it can never be completely cancelled out.”

v) The relaxation time of the magnetization has been extracted as explained in sec. V of the SI and not by simulating the a.c. experiment. The spin relaxation times, obtained in the two ways, must be exactly the same but the simulation of the a.c. experiment would be much more involved as the presence of a time-dependent magnetic field would require the numerical integration of the Redfield equation instead of an analytic one. The protocol used here is to be considered more efficient and less prone to errors than the one that simulates the a.c. experiment.

Referee: What parameters have been used for the construction of Figure 1(c)? From the inset I deduce a phonon frequency of 27.5 cm⁻¹. Having identified the model parameters used to construct Figure 1(c) one can also avoid ”arbitrary units” and put real computed spin relaxation times.

Reply: The Fig. 1 is just a cartoon to help visualizing the effects discussed in the paper and qualitatively explaining them. Moreover, the time units are indeed arbitrary as they depend on the value of a spin-phonon coupling tensor. Therefore, we prefer not to provide the details of the plot in order to avoid confusion to the readers.

Referee: Temperature dependence of the spin-relaxation time for Fe(tpaph) complex can be deduced from the maxima in the variable-frequency out-of-phase ac susceptibility, Figure S6 of Ref.23. It would be of help for the sake of comparison having experimental relaxation times along with the computed ones in one plot (Figure 3, main text).

Reply/Changes to the Manuscript: Fig. 3 has been updated to introduce the experimental values and the sentence

”The green triangles and the green line represent the experimental results as taken from ref. [24].”

has been added to the caption.

III. REVIEWER 3

Referee: The paper by Sanvito and co-workers detail the spin-phonon coupling for the case of non-discrete phonon density of states. In particular the paper addresses the importance of this case for rationalizing thermally assisted relaxation processes for molecular magnets with apparent barriers below the spectroscopically determined barriers. This contribution is very important. It is also very timely, as it clarifies a field, which recently has become increasingly messy by uncritical reporting of barriers for thermally assisted relaxation of magnetization without attention to the underlying physics. Considering the fairly heavy theoretical core material in the paper, the exposition is very pedagogical and successfully targets the chemistry-based molecular magnetism community. However, I think that several readers, like myself, will have some difficulty in figuring out if the phonon dispersion necessarily has to originate in anharmonicity. Or put in another way: is "anharmonic phonons..." in the text and title really the optimal wording if chemists should get the full message. Could "phonon dispersion" or "continuous phonon spectra.." substitute for the anharmonicity. I think this point should be discussed briefly no matter the answer. In doing this, the title should also be reconsidered.

Reply: We thank the referee for her/his kind consideration and for the appreciation expressed towards our work. In the proposed model only the gamma point phonons have been considered and any density of states broadening originates from anharmonic contributions. These result in a finite line-width to the vibrational spectra. The introduction of more details in the model, e.g. the sampling of the reciprocal space of the crystal lattice, will be tackled in future papers and it is not discussed here (this requires a significantly larger volume of calculations).

Changes to the manuscript: In order to make the point clearer to the reader, we have added the following sentence to the main text:

"It should be stressed that the effects due to the phonon line-width broadening induced by the phonon-phonon interactions are effective regardless the nature of the density of states of the lattice vibrations. Indeed, even a single phonon interacting with the spin is sufficient to induce the Ueff reduction. However, the broadening of the phonons' spectral shape also enables more phonons, close in energy, to be operative at the same time, making the vibrational density of states another important figure of merit for the interpretation of the spin-dynamics."

Referee: I was a bit surprised to find that one of the most closely related papers (Luis et. al. Nature Comm. 5: 4300) discussing phonon-assisted magnetic relaxation processes - albeit in a more narrow context - is not referenced or placed into the context of the present contribution at all. I think this should be done. It will detract absolutely nothing from the beautiful presentation in question, but will probably aid a large fraction of the readers.

Reply: We thank the referee to bring our attention to the Luis et. al. paper. This work represents an important contribution to the field and it has been included in our citations together to other contributions of similar importance.

Changes to the manuscript: We have added the following comment in the main text:

"Our finding are in line with the recent research trend that see the engineering of SMM/phonon spectra as a major challenge for the design of the next generation magnetic molecular materials [8], [31], [35], [36]."

The citation [35], [36] and [37] has been introduced in the last version of the manuscript and they corresponds to: [35] S. Gomez-Coca, A. Urtizberea, E. Cremades, P.J. Alonso, A. Camon, E. Ruiz, and F. Luis, "Origin of slow magnetic relaxation in Kramers ions with non-uniaxial anisotropy," Nat. Commun. 5, 4300–4308 (2014). [36] L. Escalera Moreno, N. Suaud, A. Gaita Arino, "Theoretical determination of the spin-vibration coupling in the highly coherent molecular spin qubit [Cu(mnt)₂]2− ." arXiv:1512.05690 (2016). [37] M. Ganzhorn, S. Klyatskaya, M. Ruben, and W. Wernsdorfer, "Strong spin-orbit coupling between a single-molecule magnet and a carbon nanotube nanoelectromechanical system.," Nat. Nanotechnol. 8, 165-169 (2012).

Referee:The paper illustrates the theoretical description nicely by application to a concrete example, [tpsPhFe]-. However, the reader is presented with absolutely no arguments for this particular choice of test system. I think a rationale for this choice would strengthen the paper and avoid speculations that the system was chosen based on agreement with the modeling.

Reply: We thank the referee for her/his suggestion. We agree that a more specific discussion on this point could

strengthen our motivations.

Changes to the manuscript: We have added the following comment in the main text:

"This specific SMM has been chosen due to its well well experimentally characterized structural and magnetic properties. The relaxation features showed by $[(tpa^{Ph})Fe]^-$ are rather general and, given the large magnetic anisotropy ($D=-27.5 \text{ cm}^{-1}$ [24]), they readily translate to most appealing single ion SMMs, comprising either a transition metal or a lanthanide ion. Moreover, this SMM has one of the most compact unit cell in the SMMs family, which makes it easier the implementation of our demanding computational framework."

Referee: In line 204: "phonons" or "phonon" ? Also the sentence commencing on line 445 is rather inelegant due to the insistence on the plural. I think the sentence ending on line 268 deserves at least one reference.

Reply: At line 204 the plural is correct. Indeed, in our $S=1$ model there is only one phonon directly coupled with the spin but at the same time this phonon is connected to an ensemble of other phonons (the phonon thermal bath). The latter ones are however not directly coupled with the spin.

Changes to the manuscript: In agreement with the referee's suggestion we have modified the sentence at line 268 in:

"...Here we have shown that the vibrational features of SMMs are the central figures of merit in the design of new systems. Indeed, frequency magnitude and life-time of phonons, along with the spin-phonon coupling coefficients, are all... "

Referee: I think the sentence ending on line 268 deserves at least one reference.

Changes to the manuscript: According to the referee suggestion we added the following statement and citation:

"Indeed, a typical experimental EPR line-width value for SMMs is of the order of 1 cm^{-1} or less [24]."

Referee: Several other recent contributions have commented on the importance of controlling/tailoring the phonon spectra of SMM crystals, these ought to be referenced in the concluding part (line 442 and onward).

Reply: Please refers to the previous point concerning the paper of Luis et al.

REVIEWERS' COMMENTS:

Reviewer #1 (Remarks to the Author):

I would like to thank the authors for carefully considering all questions and comments. I am fully satisfied with their answers and can recommend publication.

This paper clarifies our understanding of the magnetic relaxation behavior of SMM, in particular it points out the importance of an-harmonic broadened phonon for the under barrier relaxation process.

Reviewer #2 (Remarks to the Author):

In the revised version of this manuscript which I recommended for publication in Nature Communication the authors addressed all queries of this reviewer and made corresponding changes in main text and in the supporting information. In one more comment which the reviewer submitted afterwards it was proposed to include in the supporting information some more details regarding the interfacing of the CP2K program for atomistic simulations of condensed matter systems with the quantum chemical molecular code ORCA. My proposal was formulated as reproduced below, but presumably was not delivered to the authors, because this proposal was not taken into account in the revised version, neither in the supporting information nor in the response of the authors to the reviewer. This is somewhat more technical issue, which should not delay publication of this work, however. Nevertheless, I consider it important for those who will possibly appear as potential users of the method described in this work. For this reason, I recommend publication of this work without further changes, or after including the additional information, I looked for, depending on the authors response.

Here the additional comment after submission of my report:

"The authors are the first to propose some sort of interface between the periodic program CP2K for geometry optimization, and frequency and normal modes calculations and the program ORCA for the calculations of local properties such as zero-field splitting parameters and first derivatives with respect to the lattice and molecular normal modes. It would be nice to have some sample inputs and some selected part from the outputs using these two program in combination. This could be also important in some other aspects and areas for research of materials."

M Atanasov

Reviewer #3 (Remarks to the Author):

The manuscript has been revised with detailed attention to the comments made by the reviewers. I recommend publication in its present form.

I. REVIEWER 1

Referee: I would like to thank the authors for carefully considering all questions and comments. I am fully satisfied with their answers and can recommend publication. This paper clarifies our understanding of the magnetic relaxation behavior of SMM, in particular it points out the importance of an-harmonic broadened phonon for the under barrier relaxation process.

Reply: We thank again the referee for her/his positive comments about our work.

II. REVIEWER 2

Referee: In the revised version of this manuscript which I recommended for publication in Nature Communication the authors addressed all queries of this reviewer and made corresponding changes in main text and in the supporting information. In one more comment which the reviewer submitted afterwards it was proposed to include in the supporting information some more details regarding the interfacing of the CP2K program for atomistic simulations of condensed matter systems with the quantum chemical molecular code ORCA. My proposal was formulated as reproduced below, but presumably was not delivered to the authors, because this proposal was not taken into account in the revised version, neither in the supporting information nor in the response of the authors to the reviewer. This is somewhat more technical issue, which should not delay publication of this work, however. Nevertheless, I consider it important for those who will possibly appear as potential users of the method described in this work. For this reason, I recommend publication of this work without further changes, or after including the additional information, I looked for, depending on the authors response.

Here the additional comment after submission of my report:

”The authors are the first to propose some sort of interface between the periodic program CP2K for geometry optimization, and frequency and normal modes calculations and the program ORCA for the calculations of local properties such as zero-field splitting parameters and first derivatives with respect to the lattice and molecular normal modes. It would be nice to have some sample inputs and some selected part from the outputs using these two program in combination. This could be also important in some other aspects and areas for research of materials.”

M Atanasov

Reply: We thank the referee for the interest showed toward our work. In order to meet his requirements we introduced a new section in the supplementary information (Supplementary Note 6) where the cp2k input used for the calculations of geometry optimization and vibrational properties has been fully reported. Hopefully, this will be of aid for new researchers to implement our first-principles framework.

III. REVIEWER 3

Referee: The manuscript has been revised with detailed attention to the comments made by the reviewers. I recommend publication in its present form.

Reply: We thank again the referee for her/his kind consideration and for the appreciation expressed towards our work.